# Leaf Functional Traits and Their Influencing Factors in Six Typical Vegetation Communities

**DOI:** 10.3390/plants13172423

**Published:** 2024-08-30

**Authors:** Yuting Xing, Shiqin Deng, Yuanyin Bai, Zhengjie Wu, Jian Luo

**Affiliations:** Key Laboratory of Forest Ecology in Xizang Plateau of Ministry of Education, National Forest Ecosystem Observation & Research Station of Linzhi Xizang, Institute of Xizang Plateau Ecology, Xizang Agricultural and Animal Husbandry University, Nyingchi 860000, China; xyutingting@163.com (Y.X.); 17785307164@163.com (S.D.); 17612931353@163.com (Y.B.); wzjdyx11@163.com (Z.W.)

**Keywords:** leaf functional traits, environmental factors, phylogenetic signals, RLQ analysis

## Abstract

Leaf functional traits (LFTs) have become a popular topic in ecological research in recent years. Here, we measured eight LFTs, namely leaf area (LA), specific leaf area (SLA), leaf thickness (LT), leaf dry matter content (LDMC), leaf carbon content (LCC), leaf nitrogen content (LNC), leaf phosphorus content (LPC), and leaf potassium content (LKC), in six typical vegetation communities (sclerophyllous evergreen broad-leaved forests, temperate evergreen coniferous forests, cold-temperate evergreen coniferous forests, alpine deciduous broad-leaved shrubs, alpine meadows, and alpine scree sparse vegetation) in the Chayu River Basin, southeastern Qinghai-Tibet Plateau. Our aim was to explore their relationships with evolutionary history and environmental factors by combining the RLQ and the fourth-corner method, and the method of testing phylogenetic signal. The results showed that (i) there were significant differences in the eight LFTs among the six vegetation communities; (ii) the *K* values of the eight LFTs were less than 1; and (iii) except for LCC, all other LFTs were more sensitive to environmental changes. Among these traits, LA was the most affected by the environmental factors, followed by LNC. It showed that the LFTs in the study were minimally influenced by phylogenetic development but significantly by environmental changes. This study further verified the ecological adaptability of plants to changes in environmental factors and provides a scientific basis for predicting the distribution and diffusion direction of plants under global change conditions.

## 1. Introduction

Plant functional traits are the morphological, physiological, and phenological characteristics of plants that have evolved and adapted to the environment over long periods [1]. They can respond to environmental changes alone or synergistically and closely link the environment, species, and ecosystem processes [2,3]. Plant functional traits can also characterize the adaptability of plants to environmental changes and the evolution and balance of plant functions [4]. At present, plant functional traits have been widely studied, and this field has developed rapidly [5]. It has expanded from the individual and species levels to the community and ecosystem levels and has extended to all aspects of ecological research [5,6].

Leaves are important for the production of organic matter. They provide energy for organisms and play an irreplaceable role in maintaining the biological activities of organisms [7]. Moreover, the leaf is the organ with the largest contact area between the plant and the external environment [8]. Leaves are sensitive to environmental changes and show strong adaptability under changing environmental conditions [9]. Therefore, leaf functional traits (LFTs) are considered important for the adaptation of plants to heterogeneous environments and are widely used in plant functional trait research [10,11].

In leaf functional traits, different leaf functional traits exhibit various adaptive strategies for resource acquisition and investment [12,13]. For example, a reduced specific leaf area (SLA) is beneficial in nutrient-poor conditions as it minimizes water loss through transpiration [14]. Leaf dry matter content (LDMC) and SLA are linked to how plants strategically utilize resources and are notably responsive to shifts in climate [15]. The interplay of leaf traits, involving trade-offs, mirrors the ecological strategies of species, offering insights into their mechanisms of response and adaptation to a changing climate. Extensive research has confirmed the substantial influence of climatic elements, particularly temperature and rainfall, on the characteristics of leaf functions [16,17]. Across the globe, there is a clear trend of leaf characteristics escalating in tandem with rising temperatures [18,19], which implies that temperature plays a pivotal role in the diversification of leaf traits [20]. Concurrently, precipitation stands out as a critical factor in shaping leaf traits, fundamentally dictating the absorption of nitrogen and phosphorus in arid regions [21]. Studies of LFTs often use vegetation types as the basis for research and focus on community types such as forest vegetation [2,22], desert vegetation [1], and meadow [23]. Some scholars have found that there are significant differences in plant leaf functional traits among vegetation types [24]. Considering this discovery, studies have shown that differences in LFTs are the result of environmental filtering and genetic characteristics [25,26].

Mountain ranges, with their substantial changes in altitude, present a remarkable chance to deepen our insight into the shifts of leaf functional traits amidst varying environmental contexts [27]. Within these regions, a variety of climatic elements that are pivotal for plant vitality are subject to swift changes across minimal geographical spans [28]. It is projected that leaf functional traits will manifest differences in response to the gradients of elevation, thus offering a lens through which to view plant ecological strategies. For example, a general trend is observed where plant stature decreases and leaf robustness increases with rising altitude, contrasting with conditions found at lower elevations [29]. Conversely, the population of flowering individuals exhibits a notable upsurge as elevation gains [29]. The discernible pattern of leaf trait variation correlating with elevation is a key ecological signature [30]. Comprehending these patterns of variation along environmental gradients is crucial for the study of community ecology. It is necessary to determine the relative contributions of phylogeny and environment on leaf trait variation to acquire a better understanding of the underlying factors influencing the variations in LFTs. In this study, we aimed to determine (1) whether the eight LFTs differ among communities; (2) the extent to which LFTs are influenced by evolutionary history; and (3) the relationships between LFTs and environmental factors, particularly which environmental factors influence LFTs. Our objectives were to reveal the adaptation strategies of plants in different communities along the Chayu River Basin in response to environmental changes, to provide a scientific basis for predicting plant geographical distribution patterns under global climate change, and to contribute to the enhancement of vegetation protection on the Qinghai-Tibet Plateau.

## 2. Results

### 2.1. Comparative Analysis of LFTs among Plant Communities

Figure 1 shows that there were significant differences in the eight LFTs among the six typical vegetation communities (*p* < 0.001). LA was largest in the sclerophyllous evergreen broad-leaved forests and smallest in the alpine scree sparse vegetation. The largest value of SLA was found in the alpine meadow and the smallest value was found in the alpine scree sparse vegetation. LT was lowest in the temperate evergreen coniferous forest and highest in the alpine scree sparse vegetation, and tended to decrease and then increase with increasing elevation. LDMC had the greatest value in temperate evergreen coniferous forests and the smallest value in alpine scree sparse vegetation. LNC differed among communities (*p* < 0.05) and was lowest in the temperate evergreen coniferous forest and highest in the alpine scree sparse vegetation, with a general trend of decreasing and then increasing with elevation in the vegetation zone. The highest LCC value was observed in the temperate evergreen coniferous forest and the lowest value was observed in the alpine meadow. LKC differed significantly (*p* < 0.05) between the temperate evergreen coniferous forest and other communities, and the LKC contents were the lowest. The highest LPC was observed in the cold-temperate evergreen coniferous forest, and the lowest LPC content was observed in the alpine deciduous broad-leaved shrub community.

### 2.2. Phylogenetic Composition of the Plant Community

The analysis of the species in the study area revealed a total of 241 species (including varieties) in 171 genera, 34 orders, and 73 families (Appendix A). A phylogenetic tree was constructed for these species, as shown in Figure 2. The dominant orders were Asterales and Rosales, accounting for 12.45% and 8.71% of the total, respectively. The dominant families were Asteraceae and Rosaceae, accounting for 11.20% and 7.88% of the total, respectively. The dominant genera were *Pedicularis* L., *Saussurea* DC., and *Lonicera* L., accounting for a total of 7.47% of the total. In the phylogenetic tree, several species of Asterales exhibited among the longest branch lengths, which may indicate a longer evolutionary history. Conversely, the branch length of *Hymenophyllum badium* Hook. & Grev. was the shortest, suggesting it may have a slower rate of evolution.

### 2.3. The Phylogenetic Signals of the LFTs

As shown in Table 1, the *K* values of the eight LFTs of the typical vegetation communities in this study area were all less than 1, indicating that the phylogenetic signals of the plant communities in this area were weak and did not show strong phylogenetic conservation and that the species converged and evolved. LT, LPC, and LKC had significant phylogenetic signals (*p* < 0.05), while LA, SLA, LDMC, LNC, and LCC did not have significant phylogenetic signals, indicating that the plant functional traits had weak relationships with genetic evolution and that environmental factors had a certain impact on interspecific variation. LIPA was used to test the local autocorrelation between species and traits and is used to identify species with strong phylogenetic signals. As shown in Figure 3, in the typical vegetation communities of the Chayu River Basin, a higher number of plant species were observed to exhibit positive autocorrelation in LKC, followed by LPC. There were 15 species with positive correlations with LKC, with Asterales and Ericales having the largest numbers of such species. In Asterales, *Cyananthus hookeri* C. B. Clarke, *Saussurea pachyneura* Franch., *Ainsliaea latifolia* (D. Don) Sch. Bip., and *Parasenecio quinquelobus* (Wall. ex DC.) Y. L. Chen were significantly positively correlated, but other Asterales showed no correlation. The same situation was observed for Ericales, Ranunculales, Malpighiales, Fagales, and Dioscoreales. The most significant positive correlation for LKC was with the species in sclerophyllous evergreen broad-leaved forests, accounting for 33.3% of the total. For LPC, the orders Asterales, Fagales, Ericales, and Dipsacales followed the same pattern observed between order Asterales and LKC. The species in temperate evergreen coniferous forests and cold-temperate evergreen coniferous forests accounted for the largest proportion of significant positive correlations with LPC, accounting for 42.9% of the total.

### 2.4. Relationships between Functional Traits and Environmental Factors

The relationships between environmental and species traits were determined by RLQ analysis combined with fourth-corner analysis. These tests were used to assess the significance of the relationships between LFTs and environmental factors. The results of the RLQ analysis combined with the fourth-corner analysis are shown in Figure 4 and Figure 5 and Appendix A. LA had a negative correlation with bio4 and bio7 (*p* < 0.05), no correlation with bio2, a positive correlation with other climatic factors (*p* < 0.05), and a negative correlation with elevation (*p* < 0.05). Positive correlations were observed between SLA and AK (*p* < 0.05); LT and bio2 and TK (*p* < 0.05); LDMC and bio2 (*p* < 0.05); LNC and bio1, bio5, bio6, bio8, bio9, bio10, and bio11 (*p* < 0.05); and LCC and all the environmental factors. Positive correlations (*p* < 0.05) were observed with bio2, bio7, total phosphorus, total potassium, and elevation, and there was no correlation between LCC and any of the monitored environmental factors. A negative correlation was observed between LPC and AN, and a positive correlation (*p* < 0.05) was observed between LKC and TP.

## 3. Discussion

### 3.1. Differences in LFTs among Plant Communities

Plants exhibit significant adaptability and plasticity by adjusting their morphological and physiological traits to alleviate environmental pressures and enhance the acquisition of limited resources. This, in turn, strengthens their adaptive capacity to the environment and increases their fitness [31,32]. Such adaptability is a direct reflection of the evolution of plant survival strategies, as they respond to various environmental factors through variations in their functional traits [33,34,35]. In this study, there were significant differences in the eight LFTs among the six typical plant communities, which is consistent with the results of previous studies [36,37]. This is because the LFTs of plants are closely related to the way the plants use resources, and under the long-term influence of environmental conditions, plants will gradually develop leaf traits that are adapted to the unique resources of a region [11].

LA was largest in the sclerophyllous evergreen broad-leaved forests and smallest in the alpine scree sparse vegetation, showing a decreasing trend with elevation, while LT first decreased and then increased with the elevation of the vegetation belt. There was a negative correlation between LA and LT in all the communities except for the sclerophyllous evergreen broad-leaved forests. Studies have shown that LA and LT are related to temperature and rainfall [38]. In environments characterized by high temperatures or abundant rainfall, plants often exhibit a greater LA, which can enhance photosynthetic capacity and water use efficiency [12]. The larger LA allows for increased light capture, which is essential for maximizing carbon gain in such favorable conditions. Moreover, the lower LT in these conditions may be associated with a need for efficient heat dissipation and higher rates of transpiration, which helps to prevent overheating [39]. Conversely, in areas with low temperatures or limited rainfall, plants tend to have a smaller LA, which can reduce the potential for water loss through transpiration, especially under drought conditions [40]. The higher LT in these environments may serve as an adaptation to reduce water loss by providing a barrier to water vapor diffusion and enhancing the structural support needed to withstand cold and dry conditions [12,41]. Additionally, a thicker leaf can be advantageous for nutrient conservation and slow nutrient cycling in nutrient-poor soils, as it allows for a longer retention of nutrients within the leaf tissue [12,42]. The plants in the sclerophyllous evergreen broad-leaved forests had the largest leaf areas and greatest thicknesses because, in the study area, this community type is located in a region with a subtropical climate [43]. In regions characterized by a subtropical climate, when the temperature is high in the summer, there is adequate precipitation. The rain and heat occur in the same period, and the duration of the rainy season is long. In such conditions, a large LA is convenient for transpiration [20]. The winter season in this region has mild temperatures and less rain, and a thicker LT can prevent water loss in such conditions. Moreover, *Quercus oxyodon*, the dominant tree species in the community, has hard and thick leaves with little pubescence, which makes the tree suitable for the habitat of the region [44].

SLA and LDMC reflect the ability of plants to obtain and utilize resources such as light, water, and nutrients [14]. In general, the photosynthetic products of plants with a lower SLA are mostly used to construct guard cells or increase the density of mesophyll cells, thus increasing the distance or resistance of water diffusion from the inside of the leaves to the surface of the leaves [45]. The LDMC increases accordingly to reduce the water loss caused by transpiration and improve water use efficiency, providing a strong ability to resist drought stress and survive in barren areas and improving the adaptability to the environment [46]. In this study, the SLA and LDMC of plants in the alpine scree sparse vegetation were lower than those of plants in other community types, but the LT in this community was greater. The highest LT observed in plants within the alpine scree sparse vegetation can be attributed to their succulent leaves and the presence of pubescence. These traits serve multiple purposes: they facilitate water retention and offer protection against powerful gusts of wind [47]. Moreover, they provide a defense mechanism against the intense solar radiation and the harsh cold temperatures typical of the region [48]. For instance, the dominant species of this ecosystem, such as *Corydalis benecincta* W. W. Sm. and *Eutrema verticillatum* (Jeffrey & W. W. Sm.) Al-Shehbaz & Warwick, are characterized by their fleshy leaves and roots, which are adaptations that enhance their resilience to the local environmental conditions. Among the six communities, alpine meadows had the highest SLA and the lowest LDMC, and the plants in this community adopted a resource acquisition strategy. This finding was consistent with the results reported by Liu et al. for alpine meadows in Gannan [49]. A higher SLA allowed plants to rapidly accumulate carbon during the short growing season to adapt to the harsh dry and cold environment of alpine meadows; alpine deciduous broad-leaved shrubs had a lower SLA and higher LDMC [50]. To maximize the retention of water and allocate more biomass to the plant mesophyll cells, plants in alpine deciduous broad-leaved shrub communities often adopt resource conservation strategies to adapt to the environment. Such strategies include reducing the potential growth rate, prolonging leaf life, and reducing nutrient utilization efficiency, to accumulate more carbohydrates in a relatively short growing season. Therefore, a deciduous habit has been adopted to cope with drought and other stresses [51,52].

The plant nutrient content is an indicator of physiological function and reflects the photosynthetic capacity and nutrient status of plants. Studies have shown that SLA and leaf nutrient concentration are directly proportional to the rate of resource uptake and growth [53,54,55]. In this study, the temperate evergreen coniferous forest had a lower SLA, and the LNC, LKC, and LPC were also relatively low in this community type; however, the LCC was relatively high. In the temperate evergreen coniferous forest, the number and activity of photosynthesis-related enzymes are reduced due to the lower contents of N, K, P, and other elements; lower CO_2_ assimilation rate; and thus lower photosynthetic efficiency of plants [56]. Moreover, the plants in the temperate evergreen coniferous forest grow slowly, and the life cycle and biomass accumulation time are relatively long in this community type; thus, the LCC content is greater, indicating that the plants in the temperate evergreen coniferous forest adopt a resource conservation strategy to adapt to the environment. Plants in the cold-temperate evergreen coniferous forests have high SLA and nutrient contents, with LPC having the highest values among the six typical vegetation communities. This finding is likely because of the high level of stand depression in cold-temperate evergreen coniferous forests, which have a dark and moist understory, and the resource acquisition strategy used by plants to adapt to this environment.

### 3.2. Relationships between Functional Traits and Phylogenetic Signals

As shown in Table 1, the *K* values of the eight LFTs of the typical vegetation communities in the Chayu River Basin were all less than 1. LT, LPC, and LKC had weak phylogenetic signals (*K* < 1, *p* < 0.05), and LA, SLA, LDMC, LNC, and LCC did not have significant phylogenetic signals (*K* < 1, *p* > 0.05). This indicates that they did not show strong phylogenetic conservatism, that functional traits are not fully consistent with evolutionary history, and that species exhibit convergent evolution. The results in Figure 3 similarly demonstrate this conclusion, showing that a greater number of species exhibit phylogenetic signals with LKC and LPC; phylogenetic consistency was not observed within orders, but consistency in functional traits within the communities was observed. This finding is consistent with the results of Liu et al., Adnan M A et al., and An et al. for the alpine meadows of the northeastern Qinghai-Tibet Plateau, plants in the arid environment of Guazhou County, Jiuquan city, Gansu Province, and seed plants in China [57,58,59]. There are many reasons for the weak phylogenetic signal. First, at the species scale, functional traits and phylogenetic relationships do not have a one-to-one correspondence, and phylogenetic relationships can be used only to indirectly judge the similarity of functional traits of the species in a community. Moreover, they do not comprehensively reflect the functional traits, so the relationship between phylogenetic information and the species and functional traits is not consistent [60]. Second, leaves are organs that are in direct contact with the environment and are sensitive to environmental changes. To adapt to their habitats, species with distant genetic relationships show certain similarities, resulting in convergence; these traits do not have an obvious phylogenetic signal [61]. The results of the studies by Song et al. and Shui et al. showed that individual leaf functional traits exhibited strong phylogenetic signals (*K* > 1, *p* < 0.05) [62,63]. This result may be due to the relatively stable environment of the community studied, lack of frequent environmental changes and biological invasion, and decreased environmental filtering. The species in this community can maintain their genetic stability and the continuity of their evolutionary history and thus show strong phylogenetic signals.

### 3.3. Response of Functional Traits to Environmental Factors

Plants enhance their adaptability to the environment by adjusting their own trait characteristics, reflecting the evolution of their survival strategies. At the regional scale, environmental filtering has a significant impact on the selection of the species pool [64,65]. The environment acts like a series of superimposed sieves, filtering and preserving species that are adapted to these conditions while excluding those that are not adapted [66]. Environmental filtering not only selects species with similar functional traits that match the environmental conditions but also encourages individuals of the same species to adapt to environmental changes by changing different trait combinations [67], leading to adaptive intraspecific variation.

LFTs reflect the adaptation strategy of plants to the environment [68]. Changes in the environment force plants to self-regulate, which directly indicates the ability of the plants to adapt to the environment [69]. As shown in Figure 5, all the LFTs except for LCC responded to environmental changes, with LA being the most sensitive to environmental changes, followed by LNC. The results showed that there was a positive correlation between LA and 16 climatic factors, indicating that temperature and water played a key role in the growth of leaves. This finding is consistent with the results of Wright et al., who suggested that leaf temperature was a key factor in the ability of plants to control their metabolic rates and that leaf size was used to regulate the temperature of the leaves themselves [20]. Leaf temperatures that were too high or too low would affect enzymes, thereby weakening the photosynthetic rate and reducing the growth rate of the plant [70,71]. In high-temperature environments, plants reduce the risk of heat damage by increasing LA and accelerating transpiration to cool their leaves [42]. In low-temperature environments, damage is reduced by reducing LA [42]. This conclusion can also explain the negative correlation between LA and bio4, bio7, and elevation found in this study. Elevation is a comprehensive factor that can directly or indirectly affect other environmental factors [72]. With increasing elevation, bio4 and bio7 increase, temperature and precipitation decrease, and leaf area decreases.

N is an essential element for plant growth, development, and metabolism and is necessary for protein and enzyme synthesis as well as material circulation and energy flow in ecosystems [73,74]. LNC can characterize the photosynthetic capacity and resource competitiveness of plants and reflect the survival strategies of plants [75]. In this study, LNC was positively correlated with elevation, bio2, bio7, TP, and TK and negatively correlated with bio1, bio5, bio6, bio8, bio9, bio10, and bio11. With increasing elevation, the temperature gradually decreased, bio2 and bio7 became larger, and the growing season became shorter. These environmental changes put greater stress on plants at high elevations and, as a result, plants in these areas require higher LNC to enhance their photosynthetic efficiency and material production capacity. These alterations allow plants to adapt to the environmental conditions found in areas with low temperatures and short growing seasons; similar findings were observed by Xu et al. [76]. Some studies have shown that there is no significant relationship between LNC and TN, which is consistent with the results of this study. However, this study also revealed that there was a positive correlation between LNC and both TP and TK. Some studies have shown that Pand K increase the cold resistance of plants. With increasing elevation, the P and K contents of plants decrease, which may be due to the plant litter and soil [77].

## 4. Materials and Methods

### 4.1. Study Area

The Chayu River Basin is located on the southeastern margin of the Qinghai-Tibet Plateau at 96°07′–97°49′ E and 27°42′–29°31′ N. It originates from the Demula Mountains. It is a double-trunk river formed by the convergence of the Sangqu and Gongrigabuqu Rivers. The total length is 295 km, and the basin area is 17,827 km^2^. It is one of the major rivers in eastern Xizang [78]. The basin is affected by the warm and humid airflow off the Indian Ocean. The climate of the area is mild and humid, with an average annual temperature of 12.1 °C, an average annual frost-free period of up to 215 days, and abundant precipitation. The average annual precipitation is more than 800 mm, creating a unique hydrothermal environment [78]. The study area’s landscape slopes from the northwest down to the southeast. It ranges from a peak height of 6844 m to a low point of 916 m, covering a significant vertical distance. As the altitude increases within the Chayu River Basin, distinct temperature zones are formed, each associated with specific plant life. In the downstream area, where the climate ranges from subtropical to tropical, the vegetation is characterized by sclerophyllous evergreen broad-leaved forests, found at altitudes between 916 and 2000 m. The midstream region, with its mountain temperate climate, is home to temperate evergreen coniferous forests, which are located between 2000 and 3400 m above sea level. Further up, the cold-temperate evergreen coniferous forests are situated between 3400 and 4200 m. The upstream area, marked by an alpine cold climate, features alpine deciduous broad-leaved shrubs between 4200 and 4500 m, alpine meadows between 4500 and 4700 m, and alpine scree sparse vegetation in the highest regions, ranging from 4700 to 6844 m (Figure 6 and Appendix A).

### 4.2. Field Investigation

The Chayu River Basin is located in the southeastern Qinghai-Tibet Plateau at the intersection of the Hengduan Mountains and the Eastern Himalayas [79]. The flora along the Chayu River transitions from tropical to temperate on the southern side of the Himalayas [80]. The area is rich in resources and retains many ancient relict plants [81]. It is the source of the differentiation and development of new alpine plants such as Rhododendron, Saxifraga, Gentiana, and Primula [82]. The basin spans a large elevation, has obvious vertical bands and various vegetation types, and is an ideal place to study the changes in LFTs among communities. Based on the investigation of seed plants in the Chayu River Basin, the typical vegetation types and their distributions in the study area are well documented [82]. To gain a better understanding of the vegetation characteristics in this area, this study was performed during the plant growth season in the Chayu River Basin in 2023. The study area showcases a distinct sequence of vegetation types that transition with changes in elevation. The vertical vegetation bands shift with elevation as follows: sclerophyllous evergreen broad-leaved forests are found from 916 to 2000 m, temperate evergreen coniferous forests are found from 2000 to 3400 m, cold-temperate evergreen coniferous forests are found from 3400 to 4200 m, alpine deciduous broad-leaved shrubs are found from 4200 to 4500 m, alpine meadows are found from 4500 to 4700 m, and alpine scree sparse vegetation are found from 4700 to 6844 m. Six typical vegetation types with obvious vegetation transitions were selected. Three sites with the same slope direction, land use intensity, and human disturbance were selected as the research plots (20 m × 20 m). Four shrub quadrats measuring 10 m × 10 m were set up in each plot, and four herb quadrats measuring 1 m × 1 m were set up at the corner of each plot. Biological information (e.g., vegetation type, individual number, coverage, height, diameter at breast height, and crown width) and geographic information (e.g., elevation, longitude, latitude, slope, and aspect) were recorded for each sample plot.

### 4.3. Acquisition of Environmental Variable Data

In each plot, the surface soil was collected via the five-point method, and the sampling depth was determined according to the observed depth of each community. Five soil samples collected in each plot were mixed, and 1000 g was transported to the laboratory [83]. The soil samples were naturally dried in the shade at room temperature, and any impurities in the soil were removed. After physical grinding, the samples were sieved through 10- and 100-mesh sieves and the chemical properties were determined by soil and agrochemical chemistry analysis methods [84]. Eight soil indicators were selected to represent the soil factors, namely soil acidity and alkalinity (pH), soil organic carbon (SOC), soil total nitrogen (TN), soil total phosphorus (TP), soil total potassium (TK), available nitrogen (AN), available phosphorus (AP), and available potassium (AK). The pH was determined by using an electrode with the Mettler FE28 instrument. SOC was determined by the potassium dichromate oxidation-external heating method, TN was determined by the Kjeldahl method, TP was determined by the molybdenum-antimony spectrophotometry method, TK and AK were determined by the flame photometric method, AN was determined by the alkali solution diffusion method, and AP was determined by the molybdenum antimony colorimetric method [85].

Historical climate data (1970–2000) were obtained from WorldClim (http://www.worldclim.org/, accessed on 3 April 2024) by downloading bioclimatic variables (Table 2) at a resolution of 30″ (approximately 1 km), and these data were sampled using ArcGIS 10.4.1 [86,87,88].

### 4.4. Measurement of Leaf Functional Traits

For each species in the sample plots, 10 individuals with good access to light, a mature reproductive stage, and a healthy appearance were selected for sampling, and leaves were sampled from four directions, namely southeast, west, north, and southwest. A total of 40 mature leaves were collected for each species. The samples were wrapped in wet paper, stored in a sealed bag, and transported back to the laboratory for the determination of functional traits [89]. In the laboratory, images were taken of the leaves, and the leaf area (LA) was calculated by ImageJ-win64 [90,91]. Leaf thickness (LT) was measured by a Vernier caliper with an accuracy of 0.01 mm. The fresh weight of wet leaves was determined via an electronic balance after removing surface water. Then, the leaves were placed in an envelope, transferred to an oven, and dried to a constant weight at 65 °C to obtain the leaf dry weight [92]. The specific leaf area (SLA) and leaf dry matter content (LDMC) were calculated. The leaf carbon content (LCC) was determined using a vario MACRO cube elemental analyzer from Elementar, Germany. The leaf nitrogen content (LNC), leaf phosphorus content (LPC), and leaf potassium content (LKC) in the soil samples were determined by the same methods used for determining the TN, TP, and TK.

### 4.5. Analysis of Differences in LFTs among Communities

Based on the field investigation and literature data, the species in the Chayu River Basin were identified, and an Excel table was used to construct tables of the plant taxa and functional characters. The data were tested for normality and homogeneity of variance in R using the shapiro.test and Bartlett.test functions from the basic R package ‘stats’ [93]. Based on the results of these tests, the Kruskal–Wallis test and Dunn test for multiple comparisons were also conducted in R 4.3.3 [69].

### 4.6. Construction of the Phylogenetic Tree

Using an Excel table, we organized the plant list from the Chayu River Basin into a standardized format of species/genus/family. This table was imported into R, where the Latin names of the species were further standardized using the U.Taxonstand package. Subsequently, we constructed a preliminary phylogenetic tree with the U.PhyloMaker package. To ensure a robust foundation for our analysis, we utilized a backbone phylogeny obtained from the megatrees GitHub repository (https://github.com/megatrees, accessed on 3 April 2024). Finally, the phylogenetic tree was refined and visualized in iTol [94,95,96].

### 4.7. Detection of Phylogenetic Signals

The phylogenetic signal (PS) is an index used to verify the relationships between functional traits and evolutionary history. The *K* value of the Brownian motion evolutionary model was used to test the phylogenetic signal strength of eight leaf functional traits of all the species in the study area, and the local indicators of phylogenetic association (LIPA) method was used to identify evolutionary branches or species groups with strong phylogenetic signals [97,98]. The phylogenetic signal *K* value and the significance (*p* value) of the LFTs were estimated in the picante package in R [99]. The importance values (*IVs*) of the species in the plots were screened to identify the representative species in the community, and the five most abundant species in the herb layer (*IV*_2_) and the shrub layer (*IV*_1_) were selected; in the tree layer, all the species were retained [85]. The Excel table was used to sort the functional trait data according to the screening results, the sorted table was imported into R, and the Phylogenetic package was used to complete the LIPA calculation and visualization [98].
(1)$${IV}_{1}({IV}_{2})=\left(RA+RC+RF\right)/3$$

*RA* is the relative abundance, *RC* is the relative cover, and *RF* is the relative frequency.

### 4.8. Analysis of the Relationships between LFTs and Environmental Factors

We delineated the relationships between Leaf Functional Traits (LFTs) and environmental factors through an integrated analytical approach, combining R-mode linked to Q-mode (RLQ) analysis with fourth-corner analysis. This comprehensive evaluation was performed using the adept ‘ade4’ package in R, which facilitated the detailed analysis and mapping of our ecological data [100].

## 5. Conclusions

Plants have evolved resource allocation patterns specific to their environment by adapting their morphological, physiological, and phenological traits. In this study, the eight LFTs were found to be significantly different among the communities studied. This is due to the unique geographical location and hydrothermal environment of the Chayu River Basin, which supports rich and diverse community types. Plant functional traits adapt to the environments in which they are found. Further research revealed that the differences in LFTs between communities were less affected by phylogenetic relationships and more affected by environmental changes. Among the LFTs, LA and LNC were more sensitive to environmental changes and had significant correlations with climatic factors and elevation but were less limited by soil factors. This study revealed the adaptation strategies of plants in different communities to environmental changes and provides a scientific basis for the prediction and protection of plant distributions under global climate change conditions.

## Figures and Tables

**Figure 1 plants-13-02423-f001:**
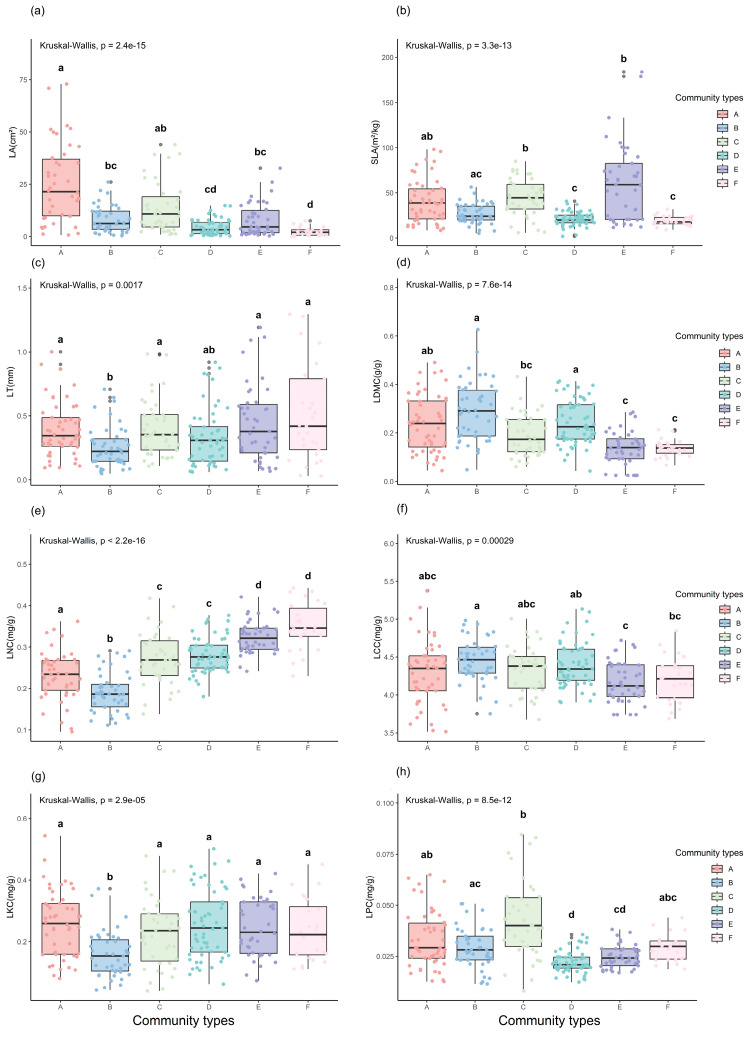
Differences in leaf functional traits among plant communities. Leaf functional traits (LFTs): (**a**) leaf area (LA), (**b**) specific leaf area (SLA), (**c**) leaf thickness (LT), (**d**) leaf dry matter content (LDMC), (**e**) leaf carbon content (LCC), (**f**) leaf nitrogen content (LNC), (**g**) leaf phosphorus content (LPC), and (**h**) leaf potassium content (LKC). Community types: sclerophyllous evergreen broad-leaved forests (A), temperate evergreen coniferous forests (B), cold-temperate evergreen coniferous forests (C), alpine deciduous broad-leaved shrubs (D), alpine meadows (E), and alpine scree sparse vegetation (F). Significant differences in post hoc Dunn tests are represented by different letters above the boxplots.

**Figure 2 plants-13-02423-f002:**
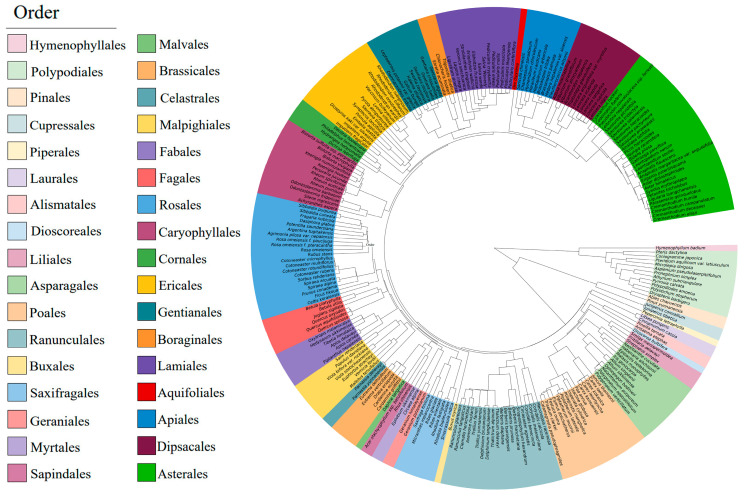
Phylogenetic tree of typical vegetation communities in the Chayu River Basin.

**Figure 3 plants-13-02423-f003:**
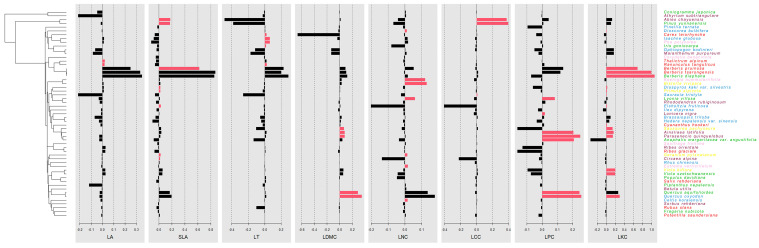
Phylogenetic correlation of species in typical vegetation communities in the Chayu River Basin. Species names are colored according to community types, blue: sclerophyllous evergreen broad-leaved forests; green: temperate evergreen coniferous forests; purple: cold-temperate evergreen coniferous forests; red: alpine deciduous broad-leaved shrubs; yellow: alpine meadows; pink: alpine scree sparse vegetation. Black bar: no positive autocorrelation; red bar: positive autocorrelation.

**Figure 4 plants-13-02423-f004:**
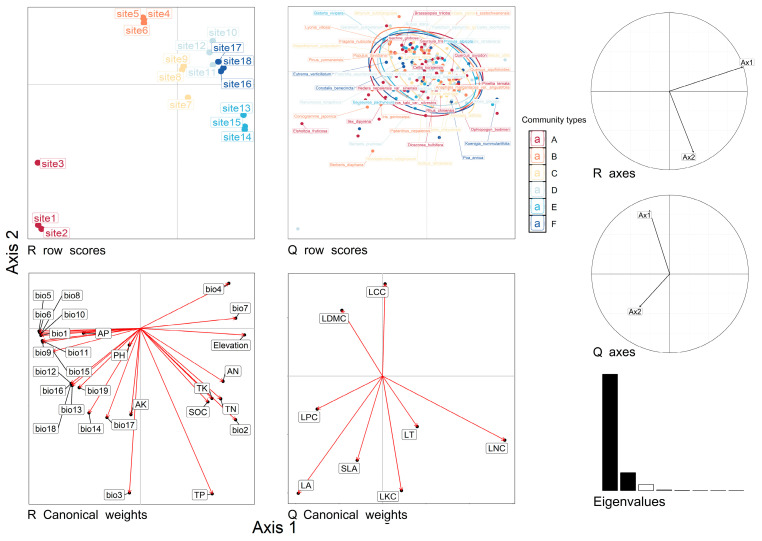
RLQ analysis of the Chayu River Basin. Study sites and species names are colored according to community types, red: sclerophyllous evergreen broad-leaved forests; orange: temperate evergreen coniferous forests; yellow: cold-temperate evergreen coniferous forests; green: alpine deciduous broad-leaved shrubs; light blue: alpine meadows; blue: alpine scree sparse vegetation.

**Figure 5 plants-13-02423-f005:**
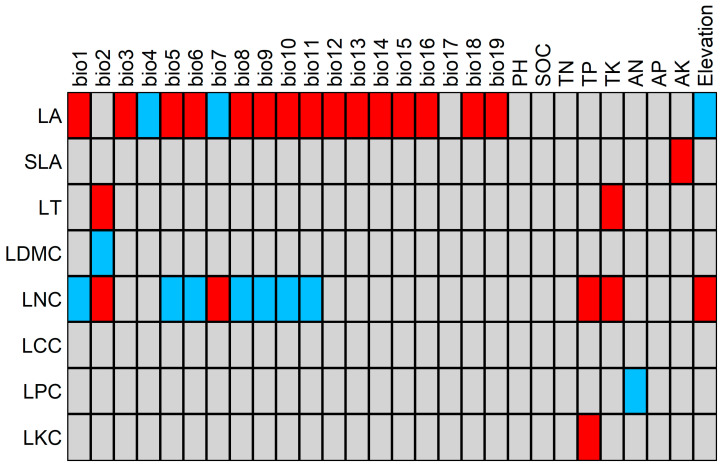
Fourth-corner analysis diagram. Red squares indicate positive relationships, blue squares indicate negative relationships, and gray squares indicate nonsignificant.

**Figure 6 plants-13-02423-f006:**
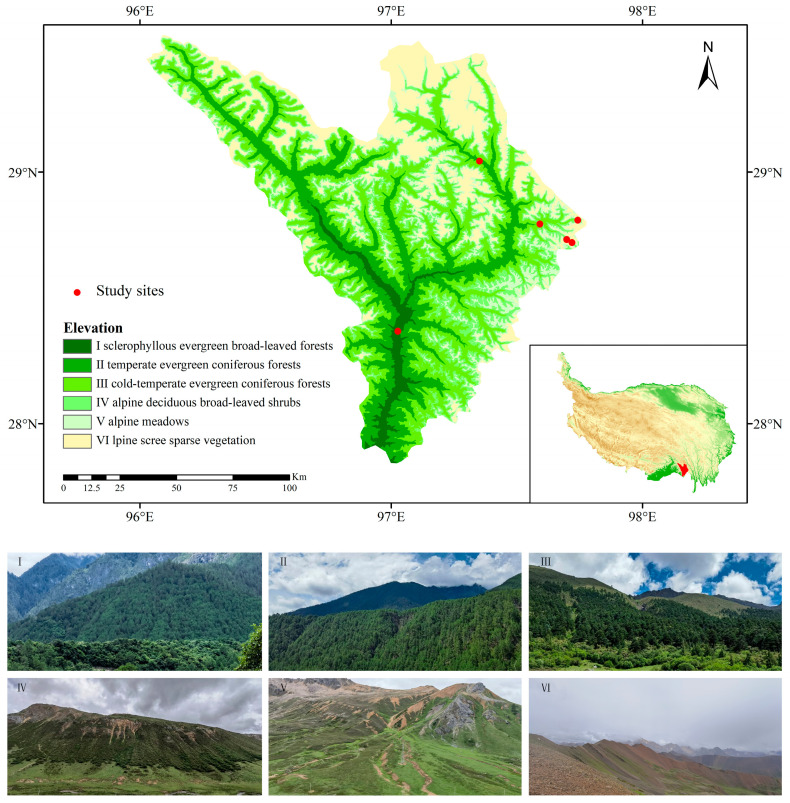
Study area and sample distribution map. Community types: sclerophyllous evergreen broad-leaved forests (**I**), temperate evergreen coniferous forests (**II**), cold-temperate evergreen coniferous forests (**III**), alpine deciduous broad-leaved shrubs (**IV**), alpine meadows (**V**), and alpine scree sparse vegetation (**VI**).

**Table 1 plants-13-02423-t001:** The phylogenetic signal of leaf functional traits.

Leaf Traits	*K*	*p*	Leaf Traits	*K*	*p*
LA	0.071137	0.243	LNC	0.041542	0.072
SLA	0.08277	0.607	LCC	0.072946	0.059
LT	0.090912	0.01 *	LPC	0.065856	0.006 *
LDMC	0.033216	0.463	LKC	0.054758	0.003 *

Significance levels (* *p* ≤ 0.05) were used in the analyses.

**Table 2 plants-13-02423-t002:** Nineteen bioclimatic variables.

Climate Variables	Description
Bio1	Annual Mean Temperature
Bio2	Mean Diurnal Range (Mean of monthly (max temp–min temp))
Bio3	Isothermality (BIO2/BIO7) (×100)
Bio4	Temperature Seasonality (standard deviation × 100)
Bio5	Max Temperature of Warmest Month
Bio6	Min Temperature of Coldest Month
Bio7	Temperature Annual Range (BIO5–BIO6)
Bio8	Mean Temperature of Wettest Quarter
Bio9	Mean Temperature of Driest Quarter
Bio10	Mean Temperature of Warmest Quarter
Bio11	Mean Temperature of Coldest Quarter
Bio12	Annual Precipitation
Bio13	Precipitation of Wettest Month
Bio14	Precipitation of Driest Month
Bio15	Precipitation Seasonality (Coefficient of Variation)
Bio16	Precipitation of Wettest Quarter
Bio17	Precipitation of Driest Quarter
Bio18	Precipitation of Warmest Quarter
Bio19	Precipitation of Coldest Quarter

## Data Availability

The original data presented in the study are openly available in Appendix A.

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
