# Peer review of "Leaf Functional Traits and Their Influencing Factors in Six Typical Vegetation Communities"

_plants, 2024, doi:10.3390/plants13172423_

Round 1

Reviewer 1 Report

Comments and Suggestions for Authors

Dear Authors

The manuscript titled "Leaf functional traits and their influencing factors in typical vegetation communities in the Chayu River Basin, southeastern Qinghai‒Tibet Plateau" provides a comprehensive analysis of the relationships between leaf functional traits (LFTs) and environmental factors across multiple vegetation communities. The study is well-structured and presents valuable insights into the ecological adaptability of plants in response to changing environmental conditions.

I have some comments and feedback enlisted here.

The abstract is too long. Kindly follow the format of the Journal.

The introduction needs more work. Kindly enlist some latest research on this area. 

Consider expanding the discussion on the potential implications of the study findings for ecosystem functioning and plant community dynamics in the Chayu River Basin. Providing contextual information on the significance of specific LFTs in terms of plant growth, resource allocation, and adaptation strategies would enhance the clarity of the manuscript. Incorporating comparisons with similar studies or literature on LFTs in alpine regions or other high-altitude ecosystems could strengthen the relevance of the research within a broader ecological context. With some enhancements to the discussion and contextualization of results, the study has the potential to make a valuable contribution to the field of plant functional trait research. 

Thanks

Comments on the Quality of English Language

Need minor edits.

Author Response

Comments :

Dear Authors

The manuscript titled "Leaf functional traits and their influencing factors in typical vegetation communities in the Chayu River Basin, southeastern Qinghai‒Tibet Plateau" provides a comprehensive analysis of the relationships between leaf functional traits (LFTs) and environmental factors across multiple vegetation communities. The study is well-structured and presents valuable insights into the ecological adaptability of plants in response to changing environmental conditions.

I have some comments and feedback enlisted here.

The abstract is too long. Kindly follow the format of the Journal.

The introduction needs more work. Kindly enlist some latest research on this area.

Consider expanding the discussion on the potential implications of the study findings for ecosystem functioning and plant community dynamics in the Chayu River Basin. Providing contextual information on the significance of specific LFTs in terms of plant growth, resource allocation, and adaptation strategies would enhance the clarity of the manuscript. Incorporating comparisons with similar studies or literature on LFTs in alpine regions or other high-altitude ecosystems could strengthen the relevance of the research within a broader ecological context. With some enhancements to the discussion and contextualization of results, the study has the potential to make a valuable contribution to the field of plant functional trait research.

Thanks

Response :

Dear Reviewer,

Thanks very much for taking your time to review this manuscript. I really appreciate all your comments and suggestions! I greatly value the feedback provided, as it is instrumental in enhancing the quality of our manuscript. Please allow me to address your suggestions point by point and to outline my revisions in the resubmitted files.

Firstly, concerning the abstract, we have revised it in accordance with the journal's guidelines, streamlining it to encapsulate the background, methods, results, and conclusions within approximately 200 words.

Moving on to the introduction section, we have supplemented it with a review of the adaptive strategies and influencing factors of leaf functional traits, which can be found from line 69 to 81. Additionally, we have included the advantages of studying leaf functional traits in mountainous regions, and this new content is presented from line 88 to 101.

Lastly, in the discussion section of our manuscript, we have added background information on the role of Leaf Functional Traits (LFTs) in plant growth, resource allocation, and adaptation strategies at lines 224-229 and 347-355. However, most of the existing research on LFTs in alpine or other high-altitude ecosystems has focused on meadows, single species, or delved into Community-Weighted Mean (CWM) values. Integrating this research into our paper is proving to be a bit challenging for me, as I am currently lacking a clear direction. I would greatly appreciate a concrete example that I could reference for guidance.

Thank you once again for your valuable comments and support. Should you require a more detailed explanation of the revisions or have any further questions, please do not hesitate to contact me.

Yours sincerely,

Yuting Xing

Reviewer 2 Report

Comments and Suggestions for Authors

Dear author(s),

Please find my comments below:

Title

1.   A new title would be better with “Leaf functional traits and their influencing factors in six typical vegetation communities”. Readers can learn where the study was conducted in M&M.

Abstract

2.   Line (L) 20, change (1) to (i).

3.   L21, remove (P<0.001).

4.   L22, remove (P<0.05).

5.   L23, remove (P<0.05).

6.   L25, change (2) to (ii).

7.   L25, explain K values?

8.   L26, change (3) to (iii).

9.   L28, check “most affected by the environment” or the most affected …

10.           L30-41, remove (P<0.05).

Keywords

11.           L47-48, the studied traits can be written as keywords instead of where the study was conducted.

Introduction

12.           L56, you can consider citing the following article at the end of the sentence as the 5th reference: https://doi.org/10.3390/ agronomy12030557

13.           L72-79, the paragraph should be removed from here to M&M.

14.           L81-84, There are two aims. Is the aim of the study different from your objective? You can merge them and write them in the abstract.

Results

15.           L91, use the abbreviation (LFTs)

16.           L121-126, the author names of the species should be written.

17.           L143-144, the author names of the species should be written.

18.            

19.            

Discussion

20.           L184-251, the subtitles could be removed.

M&M

21.           L320, explain all photos.

Conclusion

22.           .

Author Response

Dear Reviewer,

Thanks very much for taking your time to review this manuscript. I really appreciate all your comments and suggestions! I greatly value the feedback provided, as it is instrumental in enhancing the quality of our manuscript. Please allow me to address your suggestions point by point and to outline my revisions in the resubmitted files.

Thanks again!

title

Comments 1: A new title would be better with “Leaf functional traits and their influencing factors in six typical vegetation communities”. Readers can learn where the study was conducted in M&M.

Response 1:Thank you for pointing this out. We agree with this comment. Therefore, We change the title to” Leaf functional traits and their influencing factors in six typical vegetation communities”.

Abstract

Comments 2: Line (L) 20, change (1) to (i).

Response 2: Agree. We have made the changes, which can be found on line 21.

Comments 3: L21, remove (P<0.001).

Response 3: Agree. The notation (P<0.001) has been removed from line 22.

Comments 4: L22, remove (P<0.05).

Response 4: Agree. The notation (P<0.05) has been removed from line 23.

Comments 5: L23, remove (P<0.05).

Response 5: Agree. The notation (P<0.05) has been removed from line 25.

Comments 6: L25, change (2) to (ii).

Response 6: Agree. We have made the changes, which can be found on line 26.

Comments 7: L25, explain K values?

Response 7: Thank you for your question. We have removed the explanation of K values from line 27. The K value is a metric used to assess the phylogenetic signal, indicating the influence of evolutionary history on trait variation. A K value greater than 1 suggests a strong influence of phylogeny, implying convergent evolution, while a K value less than 1 indicates a weaker phylogenetic influence on the trait.

Comments 8: L26, change (3) to (iii).

Response 8: Agree. We have made the changes, which can be found on line 28.

Comments 9: L28, check “most affected by the environment” or the most affected …

Response 9: Thank you for pointing out the issue. We have revised the sentence to 'the most affected' for clarity and precision, and this change can be found on line 30.

Comments 10: L30-41, remove (P<0.05).

Response 10: In response to your comment, we have removed the notation (P<0.05) between lines 32 and 43. Additionally, due to the length of the abstract exceeding our requirements, we have also deleted the results section in that area.

Keywords

Comments 11: L47-48, the studied traits can be written as keywords instead of where the study was conducted.

Response 11: In response to the comment, we have revised the keywords section by removing the term 'Qinghai-Tibet Plateau'. This change can be found starting from line 49.

Introduction

Comments 12: L56, you can consider citing the following article at the end of the sentence as the 5th reference: https://doi.org/10.3390/ agronomy12030557

Response 12: Thank you for your recommendation. The article you suggested, which explores the 'Advantage of Multiple Pods and Compound Leaf in Kabuli Chickpea under Heat Stress Conditions' at the individual level, is indeed relevant and suitable for our review. Accordingly, we have cited this literature as the 5th reference at the end of the sentence on line 61.

Comments 13: L72-79, the paragraph should be removed from here to M&M.

Response 13:Thank you for your guidance. We have removed the paragraph from lines 102 to 109 in the introduction and relocated it to the Materials and Methods section.

Comments 14: L81-84, There are two aims. Is the aim of the study different from your objective? You can merge them and write them in the abstract.

Response 14: Acknowledging Comment 14, we have revised the manuscript accordingly. The section from lines 109 to 111, which contained a repetition of aims, has been removed. Additionally, these aims are now clearly presented in the abstract, specifically in lines 18 to 21.

Results

Comments 15: L91, use the abbreviation (LFTs)

Response 15: Acknowledging Comment 15, we have made the appropriate modification at line 122-123, where the abbreviation 'LFTs' for 'leaf functional traits' has been correctly implemented.

Comments 16: L121-126, the author names of the species should be written.

Response 16: In response to Comment 16, we have added the author names for the species at lines 153-160, as requested.

Comments 17: L143-144, the author names of the species should be written.

Response 17: Acknowledging Comment 17, we have included the author names for the species at lines 177-178.

Comments 18:

Response 18: Regarding Comment 18, I apologize for the inconvenience, but I am unable to view the content of this comment and therefore cannot provide a response.

Comments 19:

Response 19: As with the previous comment, I am unable to view this one and therefore cannot provide a response.

Discussion

Comments 20: L184-251, the subtitles could be removed.

Response 20:Thank you for your comments, but I'm not certain I understand your instruction. Does it imply that all content from lines L184 to L251 should be deleted, or is it specifically requesting the removal of the subtitle '3.1'? Could you please provide additional clarification?

M&M

Comments 21: L320, explain all photos.

Response 21: Thank you for your comments; however, I do not believe that adding explanations for the photos at L320 is the correct approach. Including explanations at L320 would create redundancy with the content already present at L422-425. Therefore, I have chosen to incorporate the photo explanations in the caption for Figure 6 instead.

Conclusion

Comments 22:

Response 22: I apologize for the inconvenience, but due to an unknown reason, I am unable to view the comment, and therefore, I cannot provide a response.

Reviewer 3 Report

Comments and Suggestions for Authors

The authors studied leaf functional traits and their influencing factors in six typical vegetation types along an altitudinal gradient in the Chayu River Basin in the southeastern Qinghai–Tibet Plateau.

Although this study is interesting and generally important, especially because of its practical significance concerning global climate change, the manuscript should be improved in order to be published.

The main objection concerns the lack of brevity of some sections, especially the Abstract and Results, as well as the insufficiently clear and poorly documented presentation of the studied sites in the Materials and Methods section. Great attention should also be paid to improving the Discussion section.

The comments are presented in more detail in the additional notes:

Abstract:

The abstract must be more concise.

1. Introduction

Lines 81-84: The aims should be summarised in a single coherent statement, and the appropriate punctuation should be used

2. Results

Line 107: For better readability,  instead of the current approach, I recommend using the compact letter display to present the Dunn test results on graphs

Lines 116-117: The complete list of taxa should be included as a supplementary table

Lines 122-124: Unnecessary explanations for the Results section

Line 165: The numerical results of these analyses should be added as supplementary tables

Line 174: The readability of Figure 4 needs to be improved

3. Discussion

Lines 195-198: The statements should be better explained, they are oversimplified and not sufficiently supported by the cited literature

Lines 200-201: Correct the sentence to make it unambiguously clear that it is not a general rule

Lines 217-219: The claim about resistance to the negative effects of strong light needs to be substantiated with literature references

Line 220: The pubescence of the leaves also improves resistance to the negative effects of strong light; link this to the previous statement about resistance to the negative effects of strong light

4. Materials and methods

In the "Materials and Methods" section, the number of sites of each vegetation type studied must be indicated in the "Study area" subsection. A list of the studied sites with their geographical location and main geographical features must be added as a supplementary table.

Lines 321-325: In view of the altitudinal gradient, the climatic characteristics cannot be generalized in this way. It is necessary to emphasize the climatic features along the altitudinal gradient.

Lines 325-326: The sentence is difficult to understand because it is too long.

Line 334: The caption for Figure 6 must include the names of the vegetation types (I - VI) that can be seen in the photos. Moreover, it  is necessary to change the map itself, three reference maps are excessive, only one is needed.

Lines 336-337: Considering that the cited source is in Chinese, it is necessary to give a slightly more detailed description of the flora of the studied area.

Line 340: It should be clarified what is meant by "obvious vegetation transitions"

Line 343: Use „corner“ instead of „apex“

Lines 357-358: Specify the type of device used to determine the pH value

Line 360: Correct „molybdenum-antimony antimony spectrophotometry“

Lines 364-366: For the WorldClim version 2.1 climate dataset for 1970-2000, cite Fick and Hijmans (2017)

Lines 381-382: Indicate the specific instrument used to determine the LCC.

Lines 389-391: Since the functions shapiro.test and Bartlett.test are part of the basic R package "stats", R itself should be cited. If both the Kruskal-Wallis and Dunn tests were performed in R, it is best to indicate that R was used for all tests, along with the appropriate source citation.

Line 396: Indicate the phylogenetic backbone used to generate the phylogenetic tree for the species in your study

Line 416: The full name of the RLQ analysis must be provided

Line 417: Specify the ade4 package correctly as indicated in the package documentation.

Author Response

Dear Reviewer,

Thanks very much for taking your time to review this manuscript. I really appreciate all your comments and suggestions! I greatly value the feedback provided, as it is instrumental in enhancing the quality of our manuscript. Please allow me to address your suggestions point by point and to outline my revisions in the resubmitted files.

Thanks again!

Abstract:

Comments 1: The abstract must be more concise.

Response 1: Thank you for your comment. In response, we have revised the abstract, streamlining it to encompass the background, methods, results, and conclusions within approximately 200 words.

  1. Introduction

Comments 2: Lines 81-84: The aims should be summarised in a single coherent statement, and the appropriate punctuation should be used

Response 2: Thank you for your comment. We have now summarized the aims in a single, coherent statement and have corrected the punctuation accordingly. These changes can be viewed at lines L111-115.

  1. Results

Comments 3: Line 107: For better readability,  instead of the current approach, I recommend using the compact letter display to present the Dunn test results on graphs

Response 3: Agree. Thank you for your suggestion to enhance readability. We have revised the presentation of the Dunn test results on the graphs according to your recommendation, opting for a compact letter display. This modification is reflected at line L138.

Comments 4: Lines 116-117: The complete list of taxa should be included as a supplementary table

Response 4: Thank you for your guidance. In response, we have included the complete list of taxa as a supplementary table. This addition can be found in the supplementary materials associated with our manuscript.

Comments 5: Lines 122-124: Unnecessary explanations for the Results section

Response 5: Thank you for your feedback. The unnecessary explanations in the Results section have been removed as per your suggestion. The revisions can be found at lines L154-157.

Comments 6: Line 165: The numerical results of these analyses should be added as supplementary tables

Response 6: Thank you very much for your suggestion. In response, we have included the numerical results of the fourth-corner analysis as supplementary tables, specifically in Table S2. You can find these at line L200.

Comments 7: Line 174: The readability of Figure 4 needs to be improved

Response 7: Thank you for your comment. We have enhanced the readability of Figure 4 by introducing distinct colors to differentiate between plots and species. Additionally, in the Q row scores graph, we have elected to display only the dominant species to prevent the overlap of species names. These modifications can be reviewed at line L209.

  1. Discussion

Comments 8: Lines 195-198: The statements should be better explained, they are oversimplified and not sufficiently supported by the cited literature

Response 8: Thank you for your comment. We appreciate the feedback regarding the simplification of the statements in lines 195-198. We have taken your comments into account and have provided a more detailed explanation in the revised manuscript. Additionally, we have incorporated supplementary literature to bolster our arguments. The revisions can be found in the updated manuscript at lines 243-255.

Comments 9: Lines 200-201: Correct the sentence to make it unambiguously clear that it is not a general rule

Response 9: Thank you for your comment on Comment 9 regarding lines 200-201. We have taken your feedback into account and modified the sentence to ensure it is unambiguously clear that the described condition is not a universal rule. The revised sentence is intended to reflect the specific context of our study area and to avoid any misinterpretation as a general principle. We have made the necessary corrections, and the updated sentence is now located at lines 258-259 in the manuscript. We believe these revisions address the concerns raised and enhance the clarity of our presentation.

Comments 10: Lines 217-219: The claim about resistance to the negative effects of strong light needs to be substantiated with literature references

Response 10: Thank you for your insightful comments and suggestions. In response to your request for substantiating the claim regarding the resistance to the negative effects of strong light, we have now included relevant literature references at line 278 of the manuscript. These references provide empirical evidence and theoretical support for the assertion made in lines 217-219.

Comments 11: Line 220: The pubescence of the leaves also improves resistance to the negative effects of strong light; link this to the previous statement about resistance to the negative effects of strong light

Response 11: Thank you for drawing our attention to the connection between the pubescence of the leaves and the resistance to the negative effects of strong light. We have revised the manuscript to clarify this relationship. We have made the necessary corrections, and the updated sentence is now located at lines 280-289 in the manuscript.

  1. Materials and methods

Comments 12: In the "Materials and Methods" section, the number of sites of each vegetation type studied must be indicated in the "Study area" subsection. A list of the studied sites with their geographical location and main geographical features must be added as a supplementary table.

Response 12: Thank you for your constructive comments and suggestions. In response to your request for specifying the number of sites for each vegetation type studied in the "Materials and Methods" section, we have made the necessary revisions.We have now included the required information in the "Study area" subsection, detailing the number of sites for each type of vegetation. Additionally, we have prepared a supplementary table that lists the studied sites along with their geographical locations and main geographical features.The modifications have been made at line 418, where we have incorporated the reference to the supplementary table.

Comments 13: Lines 321-325: In view of the altitudinal gradient, the climatic characteristics cannot be generalized in this way. It is necessary to emphasize the climatic features along the altitudinal gradient.

Response 13: Thank you for your insightful suggestion. In response to your feedback regarding the generalization of climatic characteristics along the altitudinal gradient, we have made the necessary revisions. We have now described the climatic zone types and corresponding vegetation types at different altitudes. These detailed descriptions can be found at lines 403-413, where we emphasize the specific climatic features associated with the altitudinal gradient.

Comments 14: Lines 325-326: The sentence is difficult to understand because it is too long.

Response 14: Thank you for your feedback on the clarity of our manuscript. In response to your comment about the lengthy and complex sentence at lines 325-326, we have revised the text for enhanced readability.The sentence has now been split into two separate sentences, which can be found at lines 403-404.

Comments 15: Line 334: The caption for Figure 6 must include the names of the vegetation types (I - VI) that can be seen in the photos. Moreover, it  is necessary to change the map itself, three reference maps are excessive, only one is needed.

Response 15: Thank you for your detailed comments and suggestions for improvement. We have taken your guidance on board regarding the caption for Figure 6. In accordance with your recommendation, we have revised the caption to include the names of the vegetation types (I - VI) depicted in the photographs. This addition aims to provide clearer context and enhance the reader's understanding of the figure. We have now streamlined the figure to include only one reference map, which we believe maintains the necessary information without overcomplicating the visual presentation.These revisions can be found at lines 421-425.

Comments 16: Lines 336-337: Considering that the cited source is in Chinese, it is necessary to give a slightly more detailed description of the flora of the studied area.

Response 16: Thank you for your comment on the need for a more detailed description of the flora in the studied area, especially considering the cited source is in Chinese. We have taken your suggestion into account and have expanded the description of the flora for the reader's better understanding. The revisions can now be found at lines 427-434, where we have included additional details about the plant species and their ecological significance in the area under study.

Comments 17: Line 340: It should be clarified what is meant by "obvious vegetation transitions"

Response 17: Thank you for your comment requesting clarification on the term "obvious vegetation transitions" at line 340. We have expanded upon the description of  the characteristics of the vegetation transitions in our manuscript. The clarification can now be found at lines 438-444, where we detail the distinct changes in vegetation that occur with shifts in elevation, and how these transitions are evident across our study sites.

Comments 18: Line 343: Use „corner“ instead of „apex“

Response 18: Thank you for your comment on line 343, suggesting the use of the term "corner" instead of "apex." We have made the appropriate amendment to the manuscript. The term "apex" has been replaced with "corner" to better convey the intended meaning. This change can be reviewed at line 448.

Comments 19: Lines 357-358: Specify the type of device used to determine the pH value

Response 19: Thank you for your attention to detail. In response to your comment on lines 357-358, we have now specified the type of device used for determining the pH value in our study. The revised text can be found at line 463, where we have clarified that a "Mettler FE28 pH meter" was utilized for our pH measurements.

Comments 20: Line 360: Correct „molybdenum-antimony antimony spectrophotometry“

Response 20: We acknowledge your comment on the incorrect use of the term "molybdenum-antimony antimony spectrophotometry" in our manuscript. We have reviewed the terminology and made the appropriate correction. The corrected statement is as follows: "Total phosphorus (TP) was determined by the molybdenum-antimony spectrophotometry method." This revision can be found at line 466 of our manuscript, where we have replaced the erroneous term with the correct one, ensuring the accuracy of the analytical method described.

Comments 21: Lines 364-366: For the WorldClim version 2.1 climate dataset for 1970-2000, cite Fick and Hijmans (2017)

Response 21: Thank you for your guidance on citing the WorldClim version 2.1 climate dataset for the period of 1970-2000. We have now properly cited Fick and Hijmans (2017) in our manuscript. The citation has been added to lines 472.

Comments 22: Lines 381-382: Indicate the specific instrument used to determine the LCC.

Response 22: In compliance with your request to specify the instrument used for determining the leaf carbon content (LCC), we have updated our manuscript accordingly. The specific instrument used in our study is the vario MACROcube elemental analyzer from Elementar, Germany. This information has been incorporated into the text at lines 447-448, where we now state: "The leaf carbon content (LCC) was determined using a vario MACROcube elemental analyzer from Elementar, Germany."

Comments 23: Lines 389-391: Since the functions shapiro.test and Bartlett.test are part of the basic R package "stats", R itself should be cited. If both the Kruskal-Wallis and Dunn tests were performed in R, it is best to indicate that R was used for all tests, along with the appropriate source citation.

Response 23: Thank you for your comment regarding the citation of the R package "stats" and the clarification on the use of R for statistical tests in our manuscript. We have now revised the text at lines 389-391 to accurately reflect that R was used for all statistical tests mentioned. Specifically, we have added a citation for R itself and clarified that both the Shapiro-Wilk test for normality and the Bartlett test for homogeneity of variance, as well as the Kruskal-Wallis test and Dunn test for multiple comparisons, were performed using R. This change can be reviewed at line 495-499.

Comments 24: Line 396: Indicate the phylogenetic backbone used to generate the phylogenetic tree for the species in your study

Response 24: Thank you for your comment requesting additional information on the phylogenetic backbone used in our study. We have now clarified the source of the phylogenetic backbone in the manuscript. The phylogenetic tree for the species in our study was generated using the U.PhyloMaker package, which relies on a backbone phylogeny obtained from the megatrees GitHub repository. This backbone provides a comprehensive framework that facilitates the placement and analysis of our study species within a well-supported phylogenetic context. The specific backbone used in our analysis can be found at the following GitHub link: megatrees GitHub repository. We have included this citation in the manuscript to ensure full transparency and reproducibility of our phylogenetic analysis. The revised text at line 511-513 now reads: "To ensure a robust foundation for our analysis, we utilized a backbone phylogeny ob-tained from the megatrees GitHub repository (https://github.com/megatrees)."

Comments 25: Line 416: The full name of the RLQ analysis must be provided

Response 25: Thank you for your comment on line 416, requesting the full name of the RLQ analysis. We have now included the full name of the RLQ analysis in the manuscript. This change can be reviewed at line 536.

Comments 26: Line 417: Specify the ade4 package correctly as indicated in the package documentation.

Response 26: Thank you for your comment on line 417, which highlights the need to correctly specify the ade4 package in accordance with the package documentation. We have now revised the manuscript to accurately reflect the usage of the ade4 package. The ade4 package is a powerful tool for multivariate analysis of ecological data, and it is implemented in R to perform various analyses, including the ones conducted in our study. The revised text at line 537-539 now reads: " This comprehensive evaluation was performed using the adept 'ade4' package in R, which facilitated the detailed analysis and mapping of our ecological data "